# Gain efficiency with streamlined and automated data processing: Examples from high-throughput monoclonal antibody production

Malwina Kotowicz[1], Magdalena Shumanska[1], Sven Fengler[1], Birgit Kurkowsky[1], Anja Meyer-Berhorn[1], Elisa Moretti[1], Josephine Blersch[1], Gisela Schmidt[2], Jakob Kreye[3,4,5,6], Scott van Hoof[3,5], Elisa Sánchez-Sendín[3,5], S. Momsen Reincke[3,5,6], Lars Krüger[7], Harald Prüß[3,5], Philip Denner[1], Eugenio Fava[8] & Dominik Stappert[1]*

**1** German Center for Neurodegenerative Diseases (DZNE), CRFS-LAT, Bonn, Germany, **2** German Center for Neurodegenerative Diseases (DZNE), CRFS-LIS, Bonn, Germany, **3** German Center for Neurodegenerative Diseases (DZNE) Berlin, Berlin, Germany, **4** Department of Pediatric Neurology, Charité - Universitätsmedizin Berlin, Berlin, Germany, **5** Department of Neurology and Experimental Neurology, Charité - Universitätsmedizin Berlin, Berlin, Germany, **6** Berlin Institute of Health at Charité - Universitätsmedizin Berlin, Berlin, Germany, **7** German Center for Neurodegenerative Diseases (DZNE), TTO, Bonn, Germany, **8** German Center for Neurodegenerative Diseases (DZNE), CRFS, Bonn, Germany.

\* dominik.stappert@dzne.de

## Abstract

Data management and sample tracking in complex biological workflows are essential steps to ensure necessary documentation and guarantee reusability of data and metadata. Currently, these steps pose challenges related to correct annotation and labeling, error detection, and safeguarding the quality of documentation. With growing acquisition of biological data and the expanding automatization of laboratory workflows, manual processing of sample data is no longer favorable, as it is time- and resource-consuming, prone to biases and errors, and lacks scalability and standardization. Thus, managing heterogeneous biological data calls for efficient and tailored systems, especially in laboratories run by biologists with limited computational expertise. Here, we showcase how to meet these challenges with a modular pipeline for data processing, facilitating the complex production of monoclonal antibodies from single B-cells. We present best practices for development of data processing pipelines concerned with extensive acquisition of biological data that undergoes continuous manipulation and analysis. Moreover, we assess the versatility of proposed design principles through a proof-of-concept data processing pipeline for automated induced pluripotent stem cell culture and differentiation. We show that our approach streamlines data management operations, speeds up experimental cycles and leads to enhanced reproducibility. Finally, adhering to the presented guidelines will promote compliance with FAIR principles upon publishing.

**Data availability statement:** The datasets generated and/or analysed during the current study are available in the following Zenodo repositories, https://doi.org/10.5281/zenodo.8229164 and https://zenodo.org/records/10106688.

**Funding:** We thank the Helmholtz Association for funding HIL-A03

**Competing interests:** The authors have declared that no competing interests exist.

## Introduction

Over the last few decades, technological advancements in fields such as imaging, laboratory automation, computing, and data analysis have revolutionized the way biologists work and handle data [1–4]. High-throughput (HT) and high-content (HC) studies are no longer exclusive to large, specialized labs but are gaining popularity in research conducted by smaller, independent teams [4–6]. This trend is expected to continue, as smaller biology labs increasingly adopt HT/HC techniques due to their decreasing costs, thereby generating large amounts of biological data. Additionally, the value of HT/HC techniques in producing reliable and comprehensive data has recently been highly emphasized, further incentivizing individual groups to incorporate such methods to generate high-quality research and stay competitive [7–10].

Biological data is heterogeneous by nature and often includes experimental readouts, curated annotations, and metadata, among other types of data. The increasing size and complexity of biological datasets call for effective means to manage data in its complete life cycle throughout a workflow. Generation, processing, analysis and management of heterogeneous biological data thus require tailored systems to improve data governance [11–13]. Many examples of such complex workflows, datasets and analysis processes come directly from the fields of toxicology, pharmacology and nanotechnology (e.g., vaccine development or toxicant testing), large omics studies (high-throughput generation of molecular data) and long-term (pre-) clinical studies, among others [14–17].

The increasing use of laboratory automation and the generation of experimental workflows with complex structures pose a unique challenge in backtracking and identifying samples and their related metadata. Each step of the workflow impacts the final data, and if problems arise, it can be difficult to backtrack and pinpoint errors. Likewise, the reproducibility of complex biological workflows is closely tied to precise record-keeping, especially as new techniques are introduced. As wet lab experiments are often complex, time-sensitive, and involve many researchers, the quality of documentation can be compromised. Moreover, manual data curation is time-consuming, labor-intensive, prone to human error, and at risk of biases as it relies on individual expertise. Any error in data curation compromises data integrity and can lead to incorrect conclusions, inefficient workflows, and the inability to reuse the data. Similarly, manual integration of data from multiple sources lacks standardization, has limited scalability, and can hinder early error detection [18].

To ensure data integrity in workflows and prevent potential data loss, strict quality control measures and careful monitoring of workflow steps are necessary. Although many systems exist for managing large datasets in biology [19], they are mainly implemented in larger, specialized facilities with teams trained in computer science. In smaller, individual labs, dedicated informatics staff may not be available and biologists are required to learn complex tools and technologies for data processing, despite lacking prior experience and facing time and resource constraints. Overall, there is an urgent need for design guidance for data processing solutions in biology workflows.

Here, we present a recently established pipeline for modular data processing that facilitates and documents the complex production of monoclonal antibodies (mABs) derived from individual B-cells. Implementing our data management system reduced the time-spent on data processing by over one-third and improved data reliability. Our strategy proves that, with moderate effort, biologists can set up an efficient, rewarding, systematic approach to routine data processing tasks. This approach will simplify documentation, facilitate reproducibility, and improve accuracy by eliminating errors related to manual data handling. Furthermore, data processing can be sped up, accelerating the generation of reliable insights and freeing hands for other tasks. Data processing can be standardized, enabling comparison of results across series of experiments or labs. Moreover, our approach supports scalability, as modules of data processing pipelines can be up- or down-scaled to handle varied data amounts, adapting to changing research needs.

Finally, to demonstrate the versatility and transferability of our approach, we apply it to the development of a data processing pipeline for automated stem cell culture. We show that our design guidelines can serve as best practice recommendations for other biologists and be a step towards greater reproducibility, efficiency, and standardization of workflows in biology.

> ## Box 1. Glossary
>
> **Data curation** – the process of cleaning, organizing and standardizing data towards greater quality, utility and long-term preservation. Data curation is part of data processing.
>
> **Data processing** – all tasks performed on collected data to prepare it for analysis, such as curation, formatting, transposition, joining, sub-setting and summarizing, but excluding data acquisition and storage. Data processing is a broad term and covers data transformation steps that facilitate extracting insights.
>
> **Data processing pipeline** – the sum of all data processing modules for one workflow, aimed at converting raw input data into usable information.
>
> **Data repository** – storage space to catalog and archive data. Ideally, the storage space and contained data are managed by a database software.
>
> **Knime module** – a workflow in Knime performing a series of related tasks or operations, where each operation is performed by a Knime "node" that represents the smallest operational unit of Knime.
>
> **Metadata** – data that describes other, associated data. In the context of this work, metadata includes but is not limited to: i) donor data associated with a sample (e.g., date of donation, cell number); ii) experimental settings and conditions (e.g., protocols, reagents, equipment); iii) association of experimental results with the original sample; iv) location of a given sample or its derivate.
>
> **Module or data processing module** – an individual Python script or Knime workflow.
>
> **Wet lab experiment or experimental procedure** – workflow steps that are conducted physically in the laboratory.
>
> **Workflow** – systematic series of interconnected steps designed to achieve specific research objectives, including wet lab work, data processing and data storage.

## Methodological approach, design and findings

### Data processing is key to efficient high-throughput mAB cloning and production

We have recently established a wet lab workflow that enables the production of over one thousand mABs per year from cDNA generated from patient-derived single B-cells. An antibody is composed of two protein chains, the heavy (H) and the light (L) chain, where the light chain can be of either Kappa (κ) or Lambda (λ) type. Genetic information for both the H and L chains needs to be isolated and cloned. This enables *in vitro* generation of mABs with the same specificity as in the originating B-cell. The wet lab protocols have been outlined with minor deviations elsewhere [20–22].

Briefly, the procedure consists of the following steps: cDNA is generated from individual B-cells (fluorescence-activated cell sorting (FACS) of B-cells from different originating samples, such as PBMCs or CSF, without further culture). The cDNA serves as a template to amplify the H and L chains by PCR. As the L chain can be of κ or λ type, it is necessary to perform three parallel PCR reactions (S1 Fig – PCR Heavy, Kappa, Lambda). The PCR amplicons are analyzed for correct size by electrophoresis and sequenced upon positive evaluation. Primer pairs specific for the sequenced H and L chains, and equipped with overhangs for Gibson Assembly, are selected for a final PCR reaction (for a complete primer set see Table S1). The final PCR products, covering the specificity-determining variable regions of antibodies, are cloned by Gibson Assembly into plasmids encoding the constant regions of H or L chains. The assembled plasmids are amplified in bacteria, purified and sequenced to confirm sequence identity, i.e., that no mutations causing failures on protein level have been introduced [23]. HEK cells are then transfected with pairs of plasmids encoding matching H and L chains. After transfection, mABs produced by HEK cells are secreted into the cell supernatant, collected, and their concentrations are measured.

Due to the complexity of the workflow, the production of mABs in a time- and resource-efficient manner implies considerable data processing effort. Thus, the following challenges had to be addressed for the design and standardization of our pipeline:

1. Wet lab experiments represent a selection funnel, wherein not all samples that enter the workflow proceed to the end. This is due to the involvement of a series of steps that progressively narrow down the selection of samples, i.e., not all B-cells will give rise to functional antibodies. Optimization of resource usage, therefore, requires the selection of successful samples after each analysis step. The following wet lab step is only executed when a significant number of successful samples have accumulated after a set of previous wet lab experiments (e.g., one full plate of 96 samples).

2. A specific challenge to mAB production is their composition of H and L chains. As antibodies are composed of plasmid pairs, quality analysis requires the matching of heavy and light chains cloned from the same cDNA. Only if both heavy and light chains are positively evaluated are the chains pushed to the next step. If one chain of a pair is missing, several workflow steps must be repeated for the missing chain, increasing the complexity of both data processing and documentation.

3. Another challenge to mAB production is the variability of H and L chains derived from somatic recombination of V(D)J gene segments [24,25]. The specificity-determining region of an H chain is composed of several V (variable), D (diversity), and J (joining) gene segments. The specificity-determining region of an L chain is similarly composed of several V (variable) and J (joining) gene segments. To clone H and L chains, a three-step PCR strategy is implemented (S1 Fig and S1 Table). First, forward and reverse primer mixes for each chain are used to amplify a given chain from cDNA. Then, a second PCR with primer mixes is performed to analyze the first amplicon by sequencing. Obtained sequences are analyzed for the specific V(D)J gene alleles and based on this analysis, a specific primer pair is selected from a set of 69 primers to perform a final PCR step [23]. Significant effort is required for processing the data, curating the datasets, and generating look-up tables for performing the final specific PCR.

4. The key to a time-efficient workflow is to shorten the data processing effort between two wet lab experiments. Due to the complexity of our workflow, data processing is time-consuming when done manually (≥ 33% of complete hands-on time for mABs production from cDNA). Analysis of samples, planning the next experimental step, and relating initial cDNA samples to experimental readouts are a challenge, as different machines use varying plasticware layouts (e.g., 96- or 384-well plates, column- or row-wise, individual culture plates or individual tubes).

5. Data generated on each mAB production step needs to be curated and documented before proceeding to subsequent steps, analysis, troubleshooting, hypothesis formulation, and publication.

An organized and standardized approach to data processing and analysis can help address these and similar challenges in complex biological workflows and gain efficiency. In the next sections, we present design principles that guided the development of our data processing pipeline. We start with principles that give rise to functional pipelines, such as choosing the right technologies, applying modular design and ensuring interoperability between modules. We cover the implementation of dedicated databases and design guidelines that can help develop and improve existing pipelines towards better efficiency, organization and reproducibility.

## Getting started

When designing a workflow, it is crucial to conceptualize it in advance by defining the tasks, identifying their dependencies and contingencies, and determining data processing operations. This can be done in a few steps:

1. Start by outlining a series of step-by-step experiments (tasks) that constitute the workflow, specifying data to be retrieved, analyzed and stored at each step. Determine the workflow endpoint to guide the design process. Visualizing the entire workflow helps understand the experimental sequence and anticipate potential workflow expansion (Fig 1).

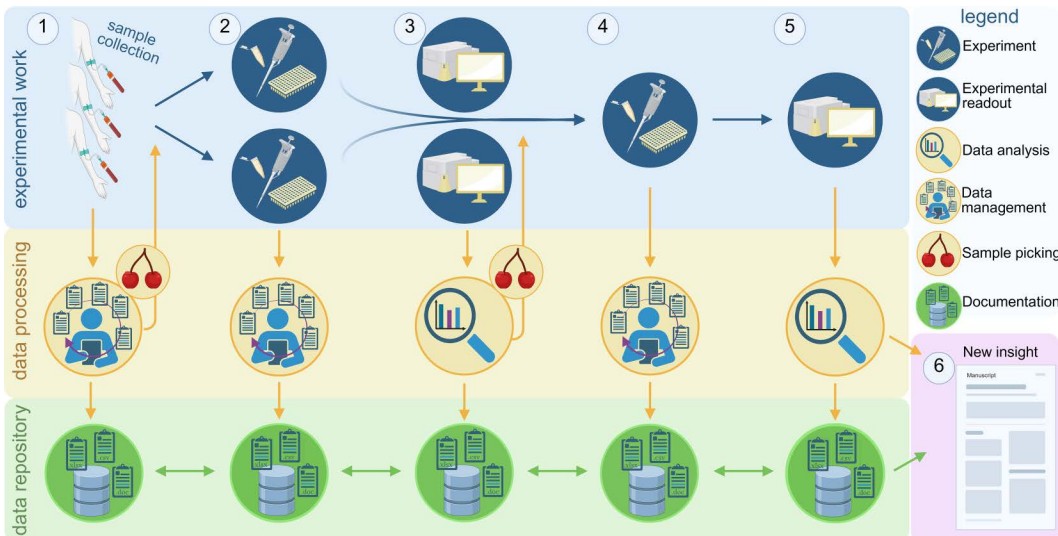

**Fig 1. An example of a two-step wet lab workflow, demonstrating data processing needs.** Data generated in wet lab experiments (experimental work, blue background, top) undergoes processing (data processing, yellow background, middle) for analysis and storage (data repository, green background, bottom). Biological samples and accompanying metadata are collected and must be curated and documented. Based on the metadata, subsets of samples are selected for analysis, such as those from wild-type or diseased subjects (1). The workflow starts with a wet lab experiment that is then documented (2) and analyzed (3). Next, the wet lab experiment is conducted on a subset of samples based on the analysis of the first experiment (4). The analysis of the second experiment closes the workflow cycle (5). Analyzing the results of the second experiment in the context of data (and metadata) gathered in the workflow cycle allows for supporting or refuting the hypotheses and yields novel insights (6).

2. Establish methods to document sample metadata and experimental details (protocols and data analysis procedures).

3. Consider repetitions that may occur in case of experiment failure and how to handle them with respect to data storage.

4. Implement ways to document the workflow executions and their results for quality control, error checking, data analysis and interpretation.

By defining these tasks, it is easier to identify the data processing operations required at each workflow step in order to construct a functional pipeline (as for example illustrated in Fig 1).

When designing the pipeline, we considered some recommendations for analysis scripts in neuroimaging already present in the literature, grounded in software development [26]. The design decisions made during development of our data processing modules offer a potentially valuable resource for other biologists in designing their own workflows. It is important to note that guidelines presented here do not always follow the sequential order of wet lab experiments. Instead, they are presented in a non-chronological manner to better explain individual pipeline operations.

We provide a link to the GitHub repository housing the data processing pipeline. This repository contains sample (mock-up) data and metadata necessary to run the pipeline steps implemented in Python. Building upon the work of van Vliet and colleagues [26], we use examples to illustrate each design principle, linking it with the scripts in the code repository (refer to the passages "Examples from the workflow"). In the sections below, we discuss the technologies used in the mAB pipeline, the module design and usage, file types used for interoperability, and our database structure.

## Choosing the right technologies

The choice of technologies for pipeline implementation is largely dependent on the skills of team members. Teams are often diverse when it comes to expertise and preferred way of working. Building upon this diversity can significantly enhance overall performance and problem-solving capabilities [27,28]. While computational skills are becoming more prevalent among biologists [20], not all researchers are proficient in programming. Software with a graphical user interface (GUI) helps users to perform complex operations by eliminating the need to learn programming. It also allows for easy, standardized and interpretable data visualization. This reduces the need to switch between different scripts or programs and decreases data processing time. GUIs can typically save and record steps taken during a certain procedure, increasing the likelihood of reproducibility among team members and minimizing errors [29]. The passage below explains our usage of certain technologies throughout the pipeline.

### Examples from the workflow

Our modules are implemented in Knime and Python, but designed in a manner to ensure interoperability (refer also to section "Defining input and output files").

Knime is among one of the most user-friendly open-source workflow management systems and offers a wide array of learning resources [4]. It provides an intuitive visual interface suitable for rapid prototyping and reproducibility. Every step in Knime is represented through an intuitive visual workflow, making it easy to document, reproduce and share processing steps and analyses. It can seamlessly connect with external tools used in biological data processing and adapt to growing datasets. Quality control steps and batch analyses can easily be automated using Knime.

Python is currently considered the most popular programming language [23], well-suited for biology applications [24]. Python has a rich ecosystem of libraries tailored to biological workflows (e.g., NumPy, SciPy, Pandas, BioPython, etc.) and a large scientific community that helps biologists solve problems faster and focus on their work, rather than spend time on complex syntax. It is also suitable for large-scale computations (e.g., GWAS) and it can run on various operating systems, further increasing usability and reproducibility.

This combination balances accessibility and customisability, making it well suited for our needs. However, other technologies can be used with similar success when designing custom pipelines (e.g., Galaxy for transcriptomics data, R-Studio, RapidMiner, etc.). The skills and preferences of team members should guide the selection of an appropriate technology solution.

**Translating tasks to modules**

Once defined, workflow tasks are converted to modules. Modules are the building blocks of a data processing pipeline, whether they be individual Python scripts or Knime workflows. By design, a module should perform a simple, singular task. To prevent scripts from becoming difficult to understand, complex tasks should be divided into smaller tasks across multiple modules. Each module should be self-contained and run with minimum dependence on other modules. This way, any changes or issues can be addressed without disrupting other modules in the pipeline.

Our pipeline follows the logic of the wet lab experiments – it is organized in modules that correspond to experimental flow on the bench (see Fig 2 for an overview of the mAB production pipeline). Since the workflow is hierarchical (each wet lab step relies on the results of the previous step(s)), we alternate wet lab experiments with data processing, rather than running the entire pipeline when all experiments are completed. Thus, the modularity of each data processing step is defined by related wet lab experiments. For example, a module can: i) analyze experimental readouts, ii) prepare layouts for the next experiment, iii) perform sample selection based on predefined criteria, iv) assemble machine-readable files for automated wet lab experiments, and v) generate a file for data storage in a database. In our experience, following the wet lab criterion is the optimal way to organize modules for hierarchical pipelines (Fig 2).

The passage below explains a typical logical flow of modules in our mAB workflow with links to sample data in GitHub.

## Examples from the workflow

To determine whether samples were successfully amplified (refer to Fig 2) in the final PCR step (*PCR3*), we perform capillary electrophoresis (*cELE2*). The results of cELE2 are processed and simplified by the module 06_cELE2.py. The module loads experimental readouts and selects a set of samples for further processing based on a band size threshold (in base pairs). The next module, 07_GiAS.py, takes the successful samples and prepares a look up table for cloning by Gibson Assembly (GiAs), as well as a database import file (Fig 2).

For reference, we provide examples of raw experimental readout data after capillary electrophoresis, look up tables (for automated liquid handling) and a database import file.

Although our workflow is semi-automated and we often deal with machine-readable files, the modules are adaptable to manual workflows. For example, the number of samples that are handled in the next workflow steps can easily be modified by defining the size of chunks in which samples are processed further. Since this workflow step operates on multiple 96-well plates, we set the number of samples to the batches of 96, but a lower number can be chosen for manual, low-throughput workflows.

**Allocating separate computational space to modules**

After designing the modules based on the mAB experimental workflow, it was important to ensure that each module runs in a separate computational space; that is, the input and output files generated by modules should be saved in dedicated directories. Ideally, script files also reside in a separate directory. This computational environment needs to be accessible

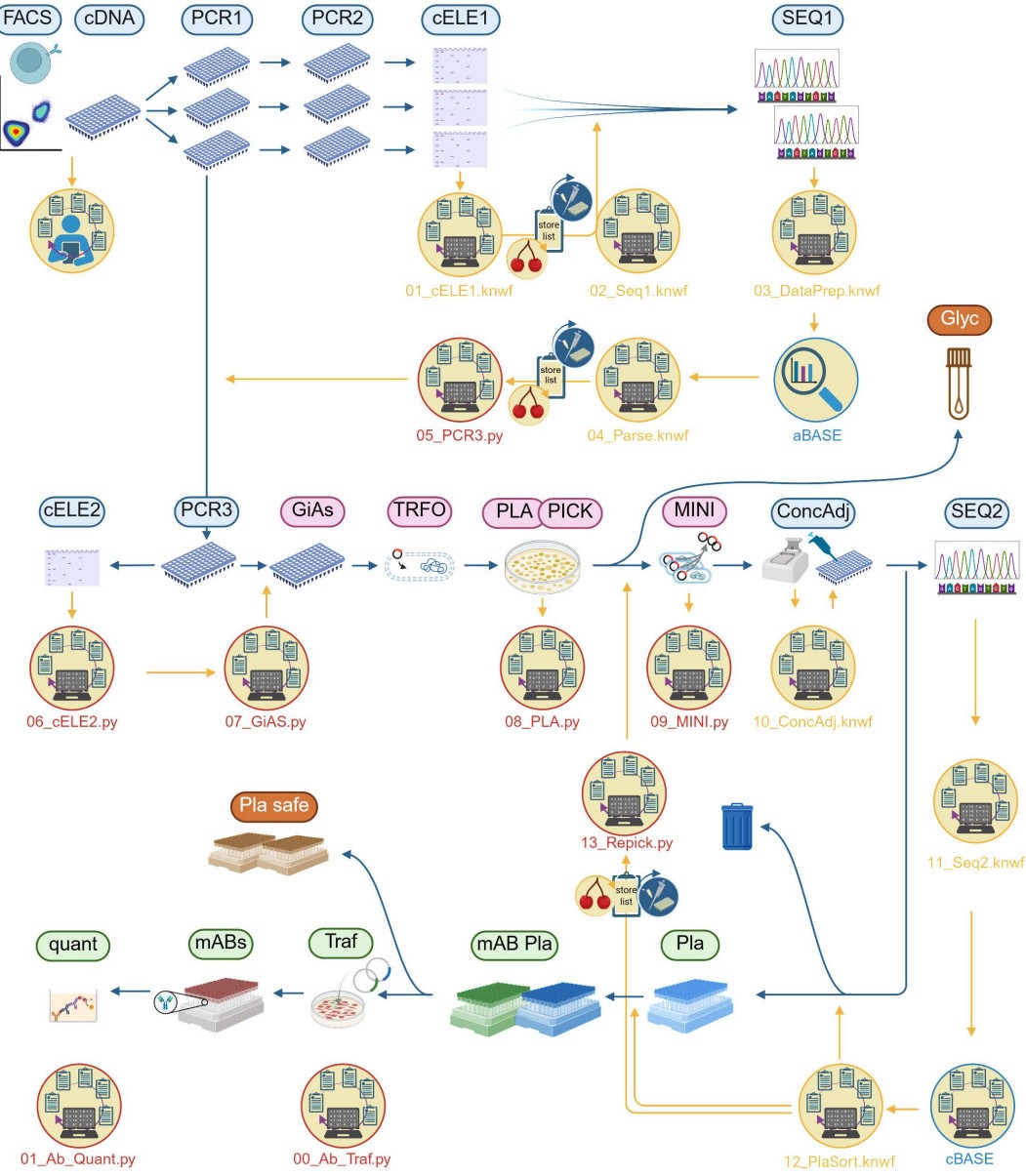

**Fig 2. Workflow for the production of mABs from patient-derived individual B-cells.** Wet lab work steps (ellipse-icons with headers) and data processing steps (yellow icons). Sample flow is depicted by blue arrows and information flow is depicted by yellow arrows. The experimental procedure starts with *FACS* sorting of individual B-cells. *cDNA* is generated from individual B-cells, and *PCR1* and *PCR2* serve to amplify antibody chains. Successfully amplified chains, evaluated by capillary electrophoresis (*cELE1*), are sequenced (*SEQ1*). Upon positive sequence evaluation with the aBASE software [23], specific primers are used for the final amplification of the specificity-determining chain parts (*PCR3*). Amplification is quality-controlled by electrophoresis (*cELE2*). The specificity-determining regions are cloned by Gibson Assembly (*GiAs*) into plasmids encoding the constant parts of the chains. Plasmids are amplified by transformed (*TRFO*) bacteria after plating (*PLA*) and picking (*PICK*) individual bacterial clones. Prior to quality control by sequencing (*SEQ2*), amplified plasmids are isolated (*MINI*), and plasmid concentration is measured and adjusted (*ConcAdj*). Based on sequencing results, functional plasmids (*PLA*) are sorted. Matching plasmid pairs (*mAB Pla*) are prepared for transfection (*Traf*) into HEK cells. mABs produced and secreted by HEK cells are harvested (*mABs*), and mAB concentration is quantified (*quant*). Glycerol stocks (*Glyc*) of transformed bacteria and plasmid aliquots (*Pla safe*) serve as safe stocks of plasmids. The colors of the wet lab work step captions indicate different workflow sections: blue – molecular biology; red – microbiology; green – cell biology. The colors of the data processing circle contours and font indicate the utilized technology: red – Python; yellow – Knime; blue – third-party software.

to all team members, must be backed up for data security, and should allow for seamless integration with different modules. These requirements are fulfilled by simple Network Attached Storage (NAS) drives, institutional set-up servers or cloud services. Within allocated computational space, the folder structure ideally reflects the architecture of the data processing pipeline.

When utilizing such storage drives and servers, it is crucial to implement appropriate data security and access control measures. This is particularly important when handling sensitive or large-scale datasets. Measures such as user authentication, role-based access, data encryption, and compliance with institutional or, if relevant, regulatory frameworks (such as GDPR or HIPAA) – for example through deidentification of patient-derived data, is essential. In our case, data access is highly project-dependent, relying on role-based access control among the team members.

Importantly, working with big datasets can pose computational challenges, especially related to memory usage and processing time. To mitigate these issues, different strategies can be implemented, such as loading data in chunks, using memory-efficient data structures (such as NumPy arrays instead of native Python lists), and leveraging parallel computing where supported. Tools like Python's multiprocessing library or KNIME's parallel execution nodes can help distribute computational load and improve performance. Although our pipeline is not affected by these issues, we recognize their relevance.

The design of our computational environment dedicated to the mAB workflow, with examples, is explained in the passage below. Links to GitHub are also provided.

## Examples from the workflow

We created a dedicated computational workspace with a folder structure that resembles our data processing steps. Keeping modules physically separated in dedicated directories allows deciding on and adhering to rigorous organization of both wet lab and data processing steps, resulting in better discipline. This easily creates access to resources (input files) required by each module without interfering with other modules. It also separates Python and Knime modules.

The published GitHub repository preserves the folder structure of Python modules in our pipeline. For example, directories 05_PCR3_Out to 09_MINI_Out are dedicated to wet lab steps from specific PCR (PCR3) to plasmid isolation (MINI), while the directory 01_Ab_Quant_Out is reserved for quantification of produced mABs (wet lab steps mABs and quant, see Fig 2). There is a separate folder for modules manipulating bacterial colony information for repicking or for input files used by modules. For reference, we also provide an image representation of an in-depth listing of directories and files for Python modules

We implemented a backup rotation scheme that involves daily (with a retention period of one week), monthly and yearly backups on an external server. To safeguard against data loss in case of the server failure, the database is also backed up on a dedicated database server twice a day.

**Defining input and output files**

Interoperability between modules (i.e., the ability of modules to work together seamlessly) should be given high priority when designing a pipeline. This ensures that data is passed between modules without loss of information or format. Moreover, each module can be developed and maintained independently, while still functioning cohesively within the larger workflow system.

To achieve this, we defined input and output files that exchange data between modules. We refer to these as *intermediate files*, as they are created as part of a larger process and are not a final output of the workflow. Intermediate files

provide standardized structure and syntax for exchanged data. They also serve as interfaces, i.e., files generated as output by one module can then be used as input by subsequent modules. The format of our input and output files is standardized. We use Comma-Separated Values (CSV) files, as they are simple and human-readable (for manual quality checks), and they are widely compatible with our physical devices used for wet lab work. Other commonly used data formats, such as JSON or XML, can also be considered, particularly when handling structured or hierarchical data from external systems or Application Programming Interfaces (APIs).

   Storing intermediate files has several advantages. When errors occur, it is possible to rerun only the modules that failed, instead of running the entire pipeline again. Furthermore, manual inspection of data is possible, which helps track the progress of the workflow and troubleshoot issues as they arise. In addition, the autonomy of each module is maintained, which decreases the complexity of the pipeline and guarantees that modules run independently without relying on data saved by other modules [26].

   The use of intermediate files in our mAB workflow (based on Fig 2) is specified in the passage below, with links to GitHub.

### Examples from the workflow

Modules in our pipeline rely on one or more intermediate files, which are generated and saved by preceding modules. Intermediate files are often used by modules, along with database export files, look up tables or experimental readouts.

For example, the module 08_PLA.py, which links the information on cloning by Gibson Assembly (GiAs, Fig 2) with transformation and plating of bacteria samples (TRFO and PLA), starts by loading the pipetting schemes (configured by using an automated colony picker) and the database export file to generate the intermediate file. The latter is taken up by the next module, 09_MINI.py, which links bacteria with isolated plasmids. The module loads the intermediate file and creates the import for the database.

Although these steps are automated in our workflow, similar logic can be applied to manual workflows. For example, the module 08_PLA.py can take up any list of samples handled in low throughput. The number of samples to be processed is automatically deducted by the module from the list of samples, and the subsequent module would infer the number of samples from the intermediate file.

Finally, intermediate files are automatically validated. For example, the number of samples is cross-checked during processing — if the expected number is not met (e.g., due to failed pivoting of split samples; this happens often when uneven number of heavy and light plasmids is parsed for transfection), an error is raised. Additionally, file headers and formats are validated against parameters specified in the configuration file, so any unexpected changes at the machine level (e.g., updated column names or formats) will trigger an error.

### Dedicated databases allow solid data documentation and efficient sample tracing

The complexity of workflows determines the demands for its documentation and sample traceability. While simple workflows that handle some dozens of samples can likely be documented using spreadsheets, complex workflows that run multistep wet lab experiments for over a hundred samples require more advanced documentation, ideally in dedicated databases. A customized database facilitates retrieval of correct information between workflow end-points and saves significant time and effort otherwise spent on sample backtracking.

   To that end, we chose FileMaker Pro Advanced (FM) as the database backend. FM is a low-code relational database management system that enables fast database creation and modification through drag-and-drop functionality [30]. Since

the database engine can be accessed through a GUI, querying data is straightforward and queries are easily modifiable. Although FM may require some initial effort to become proficient, it remains an intuitive and user-friendly tool for biologists without prior experience in database design [31–33].

FM also provides scripting functionality that we used to automate the import and export of data for further processing in downstream applications. Customizable data exports facilitate quick quality control of the samples. This, in turn, is crucial for troubleshooting and for making decisions on the sample's fate at key workflow steps.

The structure and design of our mAB-dedicated database, as well as examples, are explained in the passage below (and in S2–S4 Fig).

## Examples from the workflow

Our database was built around a concept of multiple, interconnected tables that store sets of unique records (information about sample state in the workflow). Records are connected through a series of relationships between individual tables. Their uniqueness is guaranteed by a Universal Unique Identifier – a 16-byte string assigned automatically to each record on data entry. Each subsequent related table inherits the value of the record's unique ID from a previous table, allowing the retrieval of sample information at any workflow step (S2 Fig). The data is stored in 19 distinct, interrelated tables.

Structuring the data model in multiple, interrelated tables minimizes the storage of redundant data. Information about a particular sample state is entered into the database only once and linked to other related data points as needed. This can help reducinge errors associated with redundant data entry. Additionally, it ensures data consistency across multiple experiments, as the same sample information can be reused across different projects and analyses.

For example, we keep the information of bacterial plating separate from picking. This allows picking of a new bacterial colony without the need to enter the same bacteria plate again to the database. Since bacteria plates are already linked to other information about the samples on previous wet lab steps (Fig 2, steps from *FACS* to *TRFO*), connecting a repicked colony to plating information automatically handles the connection to other data in the backend (S3 Fig).

Finally, our database provides relative flexibility, allowing the addition of new tables and seamless integration of experimental readouts. For example, the structure can be extended by adding results of functional antibody assays to the information on harvested antibodies and linking a new table through shared IDs (S4 Fig).

## Further guidance

Design decisions that extend beyond building functional pipelines and can enhance efficiency, organization and reproducibility of existing ones, are important to consider. Such include, for example, using configuration files and optimizing sample handling by processing in batches. Next, we discuss additional guidelines that can be applied to improve the performance and management of workflows beyond basic functionality.

**Configuration files to reduce repetitive code.** Rooted in software development, the DRY ("don't repeat yourself") principle implies minimizing code duplication [26]. Parameters (e.g., file or directory paths or hardware-specific parameters) are often shared between modules, and one way to make them available is to create a *configuration file* that consolidates all defined parameters in a single location. Modules that require these parameters can import them from the configuration file by the unique variable name. There are several advantages to this approach. Not only is it consistent with good programming practices, but it also reduces errors and saves time, as modifications to parameters need to be made only in the configuration file, rather than in multiple locations throughout the code [26]. To minimize the possibility

of variable duplication, we check whether a variable has already been defined in the configuration file before assigning it a value in a module. This way, we ensure that variables are used consistently, preventing unintended overwriting of data (see explanation and example in the passage below). Adhering to DRY principles makes the modules relatively resistant to modifications. All in all, using configuration files can streamline the process of adjusting parameters and settings across multiple modules.

> ### Examples from the workflow
>
> For Python modules, we provide the configuration file (config.py file) that contains file and directory paths, experimental parameters, spreadsheet metadata, and more. Each module imports only configuration parameters required for its own run. For example, module 13_Repick.py imports the paths and metadata necessary for creation of a list of bacterial colonies that are re-selected for further processing.
>
> To set up file and folder paths, we used the built-in Python *os* module. The paths are defined based on the module location when the code runs. For example, module 00_Ab_Traf.py starts by importing necessary paths from the configuration file and determines the location of folders and files relative to the current (working) directory of the module. This makes the entire pipeline independent of the operating system. Even though we usually run the code from a remote server, it can also run locally as long as the folder structure stays the same. The hierarchy and names of the directories can be adjusted by simply modifying the variables in the configuration file.
>
> Similarly, any changes made to metadata of intermediate or database files can be done in the configuration file, eliminating the need to modify in multiple locations. As an example, modifying the metadata of the files required to prepare samples for the final PCR run (PCR3; Fig 2) can easily be done by updating the configuration file. This case can be useful when changes in software occur.
>
> A similar principle applies to Knime modules. A separate configuration file specifies file and folder paths for all local machines that have access to a particular Knime workflow. A custom Knime node reads the configuration file and adapts the path as necessary.
>
> Additionally, adhering to the DRY principles, we have organized the utility functions (functions that can be used by different modules for similar tasks) into separate Python files. These files only contain the function definitions without any code. This approach allows for greater organization, reusability, and maintainability of code. The function files reside in a designated folder and are organized thematically. For example, the file microbiology_supporters.py contains the functions that assist with data processing on microbiology workflow steps, while the file plt_manipulators.py contains functions handling manipulations of the plate grids and processing samples in batches, among others.

**Efficient sample batching.** Grouping samples together for batch processing is crucial for efficient use of resources (reagents, equipment or staff). Sample batching can save the overall time and cost of the production process while reducing variability between samples and improving quality control and scalability of the workflows.

The production of mABs from individual B-cell cDNA is a multistep process, with quality assessment being performed after each wet lab step. Samples that do not meet the quality criteria are excluded from further processing, and only successful samples are selected for the next step of the workflow. Since batches of samples for each wet lab step are constantly updated, it is essential to keep track of batches already pushed forward in the workflow and of batches still waiting for their turn.

In our workflow, wet lab experiments are conducted in batches of either 96 or 384 samples. To accommodate samples that cannot be processed due to limited space on the 96- or 384-plate grid, we implemented *store lists* to keep track of the leftover samples. Store list files are updated every time new samples are advanced in the workflow, with priority given to *older* samples. This ensures that the ones that have been waiting the longest are processed first. The passage below highlights examples of how we track our samples via store lists, with links to GitHub.

---

### Examples from the workflow

As an example, the module 05_PCR3.py creates a store list that contains batches of samples for the final PCR step (PCR3; Fig 2). The module starts by loading a current store list file (updated previously by a Knime module) and automatically calculates the number of 96-sample batches that should be processed next in PCR3. Then, the batches of selected samples are pushed forward to the next workflow step and the leftover samples are saved as a new store list file.

Computing the number of sample batches automatically is beneficial, as it minimizes user input and reduces the risk of human error. Additionally, we use *argparse*, a Python library for parsing command-line arguments, to get the plate barcodes for wet lab experiments (PCR3, cELE and GiAs; Fig 2). The user inputs the latest plate numbers used on that workflow step as command-line arguments. The module parses those as arguments and automatically assigns new barcodes to current sample batches.

While our workflow is semi-automated and machine-dependent, similar principles can be applied to manual workflows. The number of samples to be processed in a single experiment can be adjusted as needed by modifying the size of a batch.

---

**Getting feedback from modules.** Because detecting errors that arise during module execution in complex workflows can be challenging, the modules should provide feedback whenever possible. Getting feedback from modules refers to collecting information from each step of the workflow to assess its performance and identify potential issues as they occur [26]. Providing feedback enables collaborative usage of the pipeline, including by team members who may not be familiar with all individual operations performed by modules.

Feedback strategies can be satisfied by user interface features within the software. Most software offers visual cues and pop-up messages to inform users about their actions. For example, FM provides import log files and notifications upon import execution. Knime allows real-time assessment of computing operations through icons, indicating whether the run was successful. The result of each node's operation can also be displayed upon request, enabling interactive troubleshooting. Another feedback strategy to consider is input validation. FM provides options of validating the IDs to be unique, non-empty and unmodifiable, and retrieves the messages on failed import if these conditions are not met. Similar validation can be implemented in Python, where incorrect user input can trigger clear and informative error messages that show what went wrong and suggest possible solutions. Additionally, a good approach to automated and recurring feedback is saving custom files that include running parameters or any other information helpful for potential troubleshooting and documentation. As a minimum requirement though, in case of scripted modules, simple printout messages help to orient the users about the status of the run. Some examples of module feedback are given in the passage below, with respective links.

---

### Examples from the workflow

We have introduced printout messages for key steps in the Python modules.

---

For example, module 05_PCR3.py provides information on the total number of samples (and 96-well plates) to process in the final PCR step, the number of leftover samples saved as a store list file, together with the directory and filename of the store list. Furthermore, the *argparse* module is used in most Python scripts. It automatically generates help, usage and error messages, which prevents incorrect input from being passed during the run.

For Knime modules, we implemented self-documenting features, where a custom file is generated during the module run. This contains metadata such as run parameters, input and output files, user data, timestamps, or messages. Together with the intermediate files saved at workflow steps, self-documentation features provide frequent feedback on the progress of data processing steps.

**Organizing the pipeline.** As a final step to maintain the pipeline organization, we recommend introducing a system that is user-friendly and not too complex, so that users are encouraged to adhere to continuous documentation. Modules that are part of the main computational pipeline should be separated from those that represent work in progress. Regular inspections and a simple cleaning strategy for modules can ensure workflow maintenance with minimal effort [25].

A manual provided for each module in the pipeline is essential to ensure proper usage and minimize the likelihood of errors due to incorrect execution. Ideally, all factors that could affect the pipeline execution should be documented. This allows for future replication of the analysis. In addition to a well-defined folder structure for hosting the modules, we recommend implementing a Version Control System (VCS) to track modifications to modules that are part of the main pipeline. Finally, adhering to a comprehensive documentation scheme aids in the maintenance of the workflow.

Writing detailed manuals, documenting module logic, and implementing version control require discipline, but contribute to efficient maintenance of complex pipelines (see examples from our workflow below). Even routine operations can require consulting the documentation. In our experience, the effort of creating manuals and adhering to version control pays off in sustaining the pipeline.

## Examples from the workflow

In our workflow, distinguishing between the main and supporting Python modules is achieved by keeping a coherent naming convention and a shared directory. Python modules have names starting with a sequential, two-digit number. The outputs of module runs are kept in separate directories, suffixed with *_Out*. Git is used as VCS to track changes to main Python modules, allowing for restoration of the previous module's versions, if needed. Manuals for each workflow step are also tracked by VCS.

We keep a documentation strategy for wet lab workflow and database imports by using Electronic Lab Notebook (ELN) software. Wet lab experiments, data imports, and modifications of database structure are recorded as separate entries in the ELN, which serves as a reliable record-keeping tool. Finally, we emphasize comprehensive code documentation, recognizing that code is read much more often than it is written [28]. This approach helps us create well-documented modules.Proof of Concept – Tailored data processing pipeline and database for automated stem cell culture

To showcase the versatility of our design, we applied the design principles to develop a data processing pipeline and a database for automated stem cell culture (ASCC). Recently, our lab has established an automated cell culture platform that integrates a robotic liquid handling workstation for cultivation and differentiation of human induced pluripotent stem cells (hiPSCs). The hiPSCs are expanded and differentiated into brain microvascular endothelial cells (BMECs)

for generation of an *in vitro* blood-brain barrier (BBB) model [34]. Mature BMECs are seeded on TransWell plates for a 2D permeability BBB model. Trans-endothelial electrical resistance (TEER) is then measured to assess the integrity of the barrier (for a detailed protocol, refer to Fengler et al. [34]). BBB models generated in high-throughput scale with close-to-physiological characteristics can facilitate the screening of BBB-penetrating drugs, aiding the development of targeted drug delivery systems for neurological disorders [35–37].

By applying our workflow design principles mentioned above, we developed a proof-of-concept pipeline focused on hiPSCs differentiation into BMECs that is flexible and can be expanded to the generation of other hiPSCs-derived cell types, including astrocytes, neurons, microglia, monocytes and more. The pipeline consists of Python scripts, FM database and FM scripts. Below, we discuss our design process and how it aligns with the already showcased design principles.

**Getting started – ASCC.**  Firstly, we started the design by outlining:

1. The wet lab experiments (cycles of thawing, seeding and harvest of cells). Refer to S5 Fig for the wet lab steps of the ASCC workflow.

2. The methods to document metadata, such as: i) hiPSCs batch information provided by a supplier; ii) culture conditions (including medium batch information and supplements); iii) the experiments' timestamps; iv) quality control data (such as cell viability assays and cell counts); v) equipment settings (parameters controlled by automation); and vi) user annotations.

3. The repetitions or pausing steps in the workflow, i.e., interruptions of the differentiation process, including freezing of cells on differentiation stages for future experiments or on expansion stages to allow differentiation into other cell types.

4. The methods of documenting the workflow for human supervision and potential error detection.

**Translating tasks to modules – ASCC.**  We then translated tasks to modules by following the wet lab criteria, making sure that each module performs a singular task and has little dependency on other modules (other than logical dependency that results from hierarchical nature of the workflow). Modules execute the following tasks: i) analyzing the experimental readouts (machine- or manually-generated), ii) parsing metadata, and iii) generating an import file for database storage.

During expansion and differentiation, iPSCs undergo seeding and harvest cycles across various plate formats (for example, 4-, 6- or 12-well plate grids). Cell count and viability assays are performed on each harvest cycle or whenever an assessment of cell confluency is required. Module 00_cellcount_parser.py reads the plate format from user's input and processes the file with cell count and viability information (generated by the automated cell viability analyzer, ViCell, Beckman Coulter). During the entire process of cultivation, cells are imaged daily for quality control. Module 01_img_parser.py parses the images generated by a confocal microscope and creates a .json file with metadata of all images linked to a given plate. As final workflow steps, cells are seeded on TransWell plates for TEER measurements. Module 02_teer_parser.py parses the experimental readouts and creates an import file for database storage.

We also implemented a configuration config.py file to store file and directory paths, spreadsheet metadata and configuration parameters. The utility functions are grouped thematically: cellcount_img_tools.py handles .txt files generated by ViCell and input_readers.py parses the user input.

**Allocating separate computational space to modules – ASCC.**  Next, we created a designated computational space for modules, input files (including metadata, experimental readouts and files exported from the database) and output files, with an automated backup rotation scheme of daily, monthly and yearly backups. We constructed designated directories for database export files for cell count and image parsing. Confocal microscopy images (per plate) and files generated by the automated cell viability analyzer are kept in individual folders. Scripts, .json files and output files generated at each workflow step also have separate directories.

**Defining input and output files – ASCC.** Further, we defined input and output files to share data between modules. Depending on the plate format and whether cells are in expansion or differentiation stage, the modules access different input files. For example, in an early expansion stage, the module 00_cellcount_parser.py takes up the database export file (automatically generated by a FM script) and the experimental readout file to generate an import file for database storage (saved with a timestamp). To facilitate the automatic imports to the database, records processed at the time are saved as separate files, updated after each script run. Modules 01_img_parser.py and 02_teer_parser.py follow similar logic when processing confocal microscopy images and TEER measurement files, respectively.

**Dedicated database – ASCC.** While conceptualizing the ASCC database structure, we considered several factors:

1. The information should not be redundant, i.e., it should only be entered in the database once. If the same information is entered more than once, there is probably a need to restructure the database and keep the redundant information in a separate table. For example, the information on the culture medium batch is registered only once and stored in a distinct table. Upon a medium change event, this information is populated by accessing the ID of the medium batch and adding it to the table storing the plate information (S6 Fig (A)).

2. The tables are organized following the thawing, seeding, harvesting, or freezing events, as they imply the initiation of a new process: e.g., change of the barcode or plate format, storing frozen cells in tanks, or collecting metadata. For example, the information on harvesting differentiated BMECs from 6-well plates by pooling and seeding on multiple 12-well TransWell plates is kept in separate tables linked through a one-to-many relationship. This ensures that any manipulations performed with the 6-well plate (medium change or cell count) are independent from those performed with the 12-well TransWell plates (TEER measurements or a well treatment). In addition, no redundant information is entered in the database, i.e., only the IDs of the 6-well plates are populated in the TransWell plate table (S6 Fig (B)).

3. We implemented separate tables also due to the different hiPSCs differentiation methods, each requiring distinct processes. Although the current ASCC database assumes BMECs differentiation, other cell types (e.g., astrocytes) will be considered in the future. In the current structure, new tables for distinct cell differentiation can easily be linked to existing ones (S7 Fig).

## Discussion

The ability to document, process and analyze large datasets to identify new patterns and relationships has become increasingly important in modern biology. In the evolving landscape of biology research, it is not only large, specialized laboratories that are embracing HT/HC techniques, but smaller biology labs are also progressively implementing similar methods. However, dealing with data generated in HT/HC schemes poses challenges related to sample backtracking, quality control, record keeping, and data curation and storage, among others. This can lead to compromised data integrity, potential data loss, and ineffective workflows. These challenges can be addressed by implementing modular data processing pipelines, which help to make substantial progress toward optimizing data management practices.

We employed the computational pipeline for data processing in a complex, semi-automated, multistep workflow for the production of mABs to tackle similar challenges and monitor all workflow steps, ultimately aiming to enhance data governance in our experimental setup. The design principles presented here help serve as guidance for the development of data processing pipelines in biology. The versatility of our approach allows for its application to diverse biological problems concerned with intensive data collection activities in a variety of settings, in which data undergoes continuous processing, analysis and modification before reaching the endpoint.

We showed how our design can help minimize the reliance on error-prone and resource-intensive manual data handling, significantly reducing errors and optimizing both time and resource usage. The modular nature of the design allows for flexibility to handle samples in high- and low-throughput settings. While the modules are designed to function in isolation, we combine them into a custom pipeline that operates on the mAB production workflow.

Our approach also helps streamline data processing and documentation, leading to faster generation of insights (allowing more time for other tasks), improving data reusability and promoting seamless data exchange. Additionally, we showed how deploying a dedicated database facilitates the standardization of data processing procedures. By implementing a tailored database, we established consistent and structured approaches for storing and organizing data, allowing us to efficiently track the samples throughout the workflow. Overall, adhering to the design principles outlined in this work can assist in enhancing the accuracy and fostering the reproducibility of data analysis by combining standardization, documentation, scalability, and error detection. Furthermore, well-structured metadata and data facilitate the utilization of machine learning and AI-based tools, which will aid quality control, process optimization and enable new analysis approaches [38].

Although our modules are tailored for mABs production, we foresee that our approach can support other biologists in building their own small-scale data processing pipelines for individual use. This approach is different from large-scale pipelines developed by bioinformaticians to manage massive volumes of data [39]. Thus, we focused our design efforts on feasibility for biologists with limited programming experience by employing software with GUI, such as Knime and FM, besides Python. With some modifications, individual modules presented here can also be adapted by other biologists and incorporated into their own workflows. To that end, we provide a GitHub repository containing mock-up data and a simplified pipeline, together with detailed instructions on how to download, install, and run it. Furthermore, we demonstrate how the code can be customized to accommodate data processing routines based on individual requirements. This can serve as a starting point for experimental researchers in constructing their pipelines by reusing and modifying the provided code.

To test the applicability of our design to other settings, we applied the design principles to automated hiPSCs culture, proving that it is possible to adapt the framework to stem cell-related research. In the long run, we expect that this approach will be beneficial for ASCC, as it can help to standardize and automate relevant data storage, and facilitate data-centric conclusions to gain novel insights. Furthermore, it can help make informed decisions for potential troubleshooting and use data to optimize processes by systematic analysis through artificial intelligence or "design of experiment" approaches. Pipelines for continuous data processing and analysis are now essential in domains such as multi-omics data integration [40, 41], HT/HC screening [42], data-driven modeling [43], as well as long-term environmental monitoring [44], evolutionary biology [45–47] or plant phenotyping [48], among others. We believe that the design recommendations proposed here can find their target audience and be a source of inspiration to other researchers in developing their own data processing modules.

In the context of data-intensive science, there is a growing demand from various stakeholders, such as the scientific community, funding agencies, publishers and industry, for data to meet the standards of being Findable, Accessible, Interoperable, and Reusable (FAIR) over the long-term. This also concerns research processes beyond data, including analytical workflows and data processing pipelines. Many projects have already adopted different elements of FAIR principles into their data (and non-data) repositories [49]. Our data processing pipelines align with FAIR principles in several ways. First, we satisfy the *findability* facet of FAIR by making the mAB and the ASCC pipelines publicly available via GitHub and assigning a persistent identifier. Clear documentation provides necessary information to understand the design decisions, data and processes involved in running the pipelines. To satisfy the *accessibility* facet, we include contact information and apply an open access policy to the code, granting unrestricted access to both the mock-up data and pipeline modules. Standardized file formats and clear naming conventions throughout the pipelines aid *interoperability* by keeping a consistent and organized approach to data representation. Moreover, using standardized data formats enables compatibility and facilitates data exchange across pipelines. To account for the *reusability* aspect of FAIR, we design modular pipelines, emphasizing the granularity of the modules. Published documentation defines how modules work and how they interact with input and output data. Finally, adhering to the outlined principles for data processing may foster compliance with FAIR principles during publication. This approach ensures that both data and metadata are well-structured, minimizing the effort to achieve data accessibility and findability upon publication.

## Limitations

One limitation of our system is partial reliance on licensed software, such as FM and GitLab. Nevertheless, open-source alternatives can be employed instead. For example, LibreOffice Base (LibreOffice Base, LibreOffice The Document Foundation) is an open-source, no-code database management software and offers similar functionality to that of FM, making it a suitable option for researchers with little to no programming experience. Alternatively, if team members have some knowledge of SQL, they can consider alternatives like MySQL (MySQL, ORACLE) for building and managing databases. Another interesting alternative to transform existing, code-based databases into interactive applications is NocoDB (NocoDB, NocoDB) – an open-source, no-code platform that turns databases into spreadsheets with intuitive interfaces, making it possible for teams to create no-code applications. It supports MySQL, SQLite or PostgreSQL databases, among others, but it also provides the functionality of building databases from scratch. Although we use GitLab (GitLab, GitLab) for Enterprise as a VCS to host the pipeline, the free GitLab version offers essential features for individual users and provides enough functionality to implement data processing pipelines (despite some storage and transfer limits). Alternatively, GitHub (GitHub, GitHub Inc.) can be used as a VCS, as it has traditionally been more widely recognized and utilized in the developer community due to its extensive user base and integration options.

Labs can also consider existing Laboratory Information Management Systems (LIMS) when automating the workflows to efficiently handle samples and associated data. LIMS are software applications used to streamline laboratory operations, sample tracking, data management and reporting, offering integration with various lab instruments [50]. While our system provides efficient workflow automation, it differs from LIMS. Although LIMS provide several advantages over custom-designed pipelines, investing in a full-fledged, commercial LIMS can be expensive. Unlike LIMS, which are lab-centric and require extensive customization to meet each lab's specific needs, our system is focused on streamlining specific aspects of data management and analysis, making it more flexible but less comprehensive for overall lab management. Additionally, as the goal of LIMS is to create a centralized system that manages all laboratory activities and data, specifying requirements for LIMS requires substantial effort.

Finally, our approach is project- or routine-centric rather than lab-centric, which can be considered a limitation due to the scale at which its application becomes advantageous. The maintenance and initial integration of each data processing module requires significant effort. This effort is justified only if manual data processing time is substantially reduced through repeated executions or if structured data processing is essential, such as for leveraging artificial intelligence approaches. While some of the proposed design principles are also applied in big data pipelines that run entirely *in silico*, our approach is dedicated to iterative data processing for wet lab experiment steps, and is not scalable to handling millions of data points per day.

## Conclusions

Our system introduces significant advancements over metadata handling systems tailored to specific fields, as for example neuroimaging, omics data, or microscopy data, as well as those that depend heavily on informatics expert knowledge for metadata processing. Designed with FAIR principles, our custom system optimizes workflow modules for reuse and seamless integration. It is cost-effective, can be tailored to specific needs, and allows rapid, iterative prototyping and experimenting with different solutions. The established modules can be complemented and expanded organically, adapting to the changing requirements of individual projects. Additionally, its low-code, user-friendly conception allows faster onboarding of biologists with varying technical expertise. By reducing the need for deep expert knowledge, our approach democratizes discussions and agreements on data and indeed metadata governance within project groups or labs. Furthermore, with a custom pipeline, the labs retain full control and ownership of the process, allowing for flexibility in response to changing needs and providing protective measures to the laboratory's intellectual property. Finally, our focus on standardized metadata management enhances data contextualization, reuse and accessibility, particularly to machine learning and artificial intelligence algorithms.

## Supporting information

**S1 Fig. Cloning strategy for variable antibody regions.** After cDNA generation, three parallel PCR reactions are performed to amplify the heavy chain variable region (light blue) and either the Kappa (light green) or Lambda (pink) variable region. Upon verification by sequencing, the variable regions (V) are cloned into plasmids encoding the constant part (C) of the respective antibody chain (dark blue: heavy constant part; dark green: Kappa light constant part; purple: Lambda light constant part). See Fig 2 for a comprehensive overview. For an overview on B-cell receptor and antibody variability, refer to Fig 1 by Khatri et al. [51] and Fig 1 by Mikocziova et al. [52].
(DOCX)

**S2 Fig. An excerpt from the database structure explaining its relational aspects.** Table C stores information on isolated plasmids of paired antibody chains (heavy and light), and Table D is a repository of plasmid aliquots stored as safe stocks in separate (physical) locations for future reproduction of plasmids or downstream applications. Table Ab_2 connects the information on plasmid pairs with subsequent transfection of HEK cells. Fields (i.e., columns) Working_Stock_ID, Safe_Stock_ID and Transfection_ID store unique IDs of each record per table. Fields starting with FK_ are records' IDs inherited from related tables. Records (sample states) are connected through one-to-many relationships in the database structure; for example, a plasmid from Table C can be used for creation of a safe stock aliquot (in Table D) multiple times, while one aliquot of safe stock (in Table D) is associated with exactly one plasmid from Table C. Similarly, one transfected HEK cell sample (i.e., transfection well, in Table Ab_1) is associated with exactly two plasmids (a pair of heavy and light chain plasmids, in Table C), while this plasmid pair can be used for multiple transfections.
(DOCX)

**S3 Fig. Simplified view of tables storing bacteria plate information (upper, green) and bacterial colony picking information (lower, blue).** Picked bacterial colonies can be connected to already imported bacteria plate information through the identifiers (Plating_ID/FK_Plating_ID), reducing the redundancy of stored data. The identifier Plating_ID is unique in the bacteria plating table but not in the colony picking table, allowing for picking of multiple colonies from the same bacteria plate (one-to-many relationship in the database structure). The colony picking table has its own unique identifier (Picking_ID).
(DOCX)

**S4 Fig. A hypothetical extension of database structure by table addition.** The flexibility of the database design enables smooth integration of new information (e.g., experimental readouts), which allows for efficient management of diverse datasets within the database. Here, the information on two hypothetical functional antibody assays (AB_Assay1 and AB_Assay2) are appended to the information on the harvested antibody through the AB_ID identifier, thus linking to any previous information on that sample (starting from the B-cell donor/patient).
(DOCX)

**S5 Fig. Wet lab steps of ASCC workflow: thawing, expansion and differentiation of iPSCs into BMECs in a functional BBB model.** Wet lab steps and timepoints (relative to the start of the differentiation – day 0: D0) are indicated by blue rectangles. Cycles of thawing (D-6), possible freezing (D-1, D6, D8), harvest and seeding (D-3, D8) of cells. Count and viability assays are carried out on D-3, D0 and D8. Media used at each timepoint are indicated by ellipses – gray: mTSER plus medium with/without rock inhibitor (RI); violet: Unconditioned Medium (UM); green: Endothelial Cell Medium with/without supplements (EC +/+, EC -/-, respectively). The TEER measurement timepoints (D10, D11) are indicated by a purple circle. For a detailed protocol, refer to Fengler et al. [34].
(DOCX)

**S6 Fig. Conceptualizing the ASCC database. A)** No-redundancy principle. The information on each medium batch is stored in the database only once (upper table, green headers). Whenever a media change is performed, the unique

identifier of the concerned medium batch (ID_medium_INFO) is fetched on the backend by a File Maker script and populated in the media change table (lower table, red headers). Information in columns::Medium Name,::Composition, and::Mix Date is fetched from the medium table (upper table, green headers). Information in the column::Barcode is fetched from another table (not shown). **B)** Example of the harvest event that guided the design of database structure: pulling differentiated cells from 6-well plate and seeding on 12-well TransWell plate. At this workflow stage, cells could also be cryo-stored for future experiments. Implementing a one-to-many relationship between 6-well and 12-well TransWell plates helps avoid storing redundant information.
(DOCX)

**S7 Fig. Conceptualizing the ASCC database-continued.** Possible expansion of database structure towards differentiation of other cell types (such as astrocytes, neurons, microglia, monocytes; red tables). The structure is flexible and allows appending of tables to a 6-well plate table (green) that stores information on undifferentiated cells through a one-to-many relationship. In this way, different differentiation protocols can easily be accommodated in the database.
(DOCX)

**S1 Table. Primers used for the amplification of heavy, light Kappa (κ) and light Lambda (λ) chains.** Three PCR reactions are performed per chain. *PCR1* and *PCR2* are performed with forward and reverse primer mixes. *PCR3* is performed with specific primer pairs selected from 69 individual primers. The selection is based on sequencing results of the second PCR amplification and identification of used V(D)J alleles. See the attached Table S1.xlsx file. Refer also to Fig 2.
(XLSX)

## Acknowledgments

We thank the Helmholtz Association for funding HIL-A03. All figures were created with BioRender.com

## Author contributions

**Conceptualization:** Malwina Kotowicz, Dominik Stappert.

**Data curation:** Malwina Kotowicz, Magdalena Shumanska, Sven Fengler, Birgit Kurkowsky, Anja Meyer-Berhorn, Dominik Stappert.

**Funding acquisition:** Jakob Kreye, Lars Krüger, Harald Prüß, Philip Denner, Eugenio Fava, Dominik Stappert.

**Investigation:** Malwina Kotowicz, Magdalena Shumanska, Sven Fengler, Birgit Kurkowsky, Anja Meyer-Berhorn, Dominik Stappert.

**Methodology:** Malwina Kotowicz, Dominik Stappert.

**Project administration:** Philip Denner, Eugenio Fava, Dominik Stappert.

**Resources:** Jakob Kreye, Scott van Hoof, Elisa Sánchez-Sendín, S. Momsen Reincke.

**Software:** Malwina Kotowicz, Dominik Stappert.

**Supervision:** Philip Denner, Eugenio Fava, Dominik Stappert.

**Validation:** Malwina Kotowicz, Magdalena Shumanska, Gisela Schmidt.

**Visualization:** Malwina Kotowicz, Magdalena Shumanska, Elisa Moretti, Dominik Stappert.

**Writing – original draft:** Malwina Kotowicz, Dominik Stappert.

**Writing – review & editing:** Malwina Kotowicz, Magdalena Shumanska, Josephine Blersch, Gisela Schmidt, Eugenio Fava, Dominik Stappert.

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
