## [Decision Letter · Decision Letter 0]

23 Oct 2024

Dear Dr. Stappert,

Thank you for submitting your manuscript to PLOS ONE. After careful consideration, we feel that it has merit but does not fully meet PLOS ONE’s publication criteria as it currently stands. Therefore, we invite you to submit a revised version of the manuscript that addresses the points raised during the review process.

We look forward to receiving your revised manuscript.

Kind regards,

Bhanwar Lal Puniya, Ph.D.

Academic Editor

PLOS ONE

Journal Requirements:

3. Please update your submission to use the PLOS LaTeX template. The template and more information on our requirements for LaTeX submissions can be found at http://journals.plos.org/plosone/s/latex .

4. We notice that your supplementary [figures/tables] are included in the manuscript file. Please remove them and upload them with the file type 'Supporting Information'. Please ensure that each Supporting Information file has a legend listed in the manuscript after the references list.

Additional Editor Comments (if provided):

-The authors should thoroughly address all the major comments from the reviewers and incorporate additional citations if relevant.

-One or more of the reviewers has recommended that you cite specific previously published works. Members of the editorial team have determined that the works referenced are not directly related to the submitted manuscript. As such, please note that it is not necessary or expected to cite the works requested by the reviewer.

Reviewers' comments:

Reviewer's Responses to Questions

**Comments to the Author**

1. Is the manuscript technically sound, and do the data support the conclusions?

Reviewer #1: Yes

Reviewer #2: No

Reviewer #3: Yes

2. Has the statistical analysis been performed appropriately and rigorously?

Reviewer #1: Yes

Reviewer #2: N/A

Reviewer #3: N/A

3. Have the authors made all data underlying the findings in their manuscript fully available?

Reviewer #1: Yes

Reviewer #2: Yes

Reviewer #3: Yes

4. Is the manuscript presented in an intelligible fashion and written in standard English?

Reviewer #1: Yes

Reviewer #2: No

Reviewer #3: Yes

Reviewer #1: This article is very good for publication, and the authors have put forth their best effort in writing and presenting the conclusion. However, some changes are required before its publication. I have mentioned all the points below:

I suggest authors to read and incorporate the information from the following articles and cite them:

mRNA vaccines as an armor to combat the infectious diseases. Travel Medicine and Infectious Disease 52:102550.

Zoonotic diseases in a changing climate scenario: Revisiting the interplay between environmental variables and infectious disease dynamics, Travel Medicine and Infectious Disease,

58:102694.

Nanovaccines: A game changing approach in the fight against infectious diseases. Biomedicine & Pharmacotherapy 167(2023):115597

Reviewer #2: The paper has addressed a nice problem; however, I have the below comments:

1. Accurately isolating individual B-cells from heterogeneous populations can be technically demanding. Techniques like flow cytometry or microfluidics need to be highly precise to ensure that only the target B-cell is selected.

2. Maintaining the viability of single B-cells during isolation and subsequent culture is crucial. Many methods can stress or damage cells, reducing their ability to produce antibodies. Please cite the below papers while discussing this:

"Integrative Toxicogenomics: Advancing Precision Medicine and Toxicology through Artificial Intelligence and OMICs Technology," Biomedicine & Pharmacotherapy, Elsevier, vol. 163, 114784, 2023.

“Advances in smoking related in-vitro inhalation toxicology: a perspective case of challenges and opportunities from progresses in lung-on-chip technologies,” ACS Chemical Research in Toxicology, vol. 34, pp. 1984-222, 2021.

“Emerging technologies for in vitro inhalation toxicology,” Advanced Healthcare Materials, Wiley, vol. 10, no. 18, pp. e2100633 2021.

3. Once isolated, the B-cells must be effectively expanded to produce sufficient quantities of antibodies. Achieving robust growth while preserving their functionality can be difficult.

4. Not all B-cells will produce high-affinity or functional antibodies. Identifying and selecting the right clones that produce the desired antibodies requires additional screening and validation steps.

5. The genetic material of isolated B-cells can be unstable, leading to mutations or loss of antibody specificity during culture. Ensuring genetic integrity is crucial for consistent antibody production.

6. Traditional methods of creating monoclonal antibodies often involve hybridoma technology, which adds complexity and can be less efficient compared to newer single-cell approaches.

7. Scaling up production from a single B-cell to large-scale antibody production can present logistical and technical challenges, including maintaining consistency and quality across batches.

8. Automating the entire process—from isolation to production—while maintaining high throughput and reproducibility can be complex and resource-intensive.

9. The process of characterizing the antibodies produced (e.g., affinity, specificity) can be time-consuming and requires sophisticated assays, adding to the overall complexity.

10. Finally, the paper should be thoroughly proofread and written like a scientific manuscript.

11. Please discuss the role of permeability because it plays a crucial role in the complex production of monoclonal antibodies from single B-cells: High permeability of the cell membrane is essential for the efficient exchange of nutrients and oxygen. This ensures that the isolated B-cells receive the necessary metabolic support for growth and antibody production. Please cite the below papers: “Investigating the Use of Machine Learning Models to Understand the Drugs Permeability Across Placenta,” IEEE Access, vol. 11, pp. 52726-52739, 2023.

“Micropatterned neurovascular interface to mimic the blood-brain barrier neurophysiology and micromechanical function,” Cells, MDPI, vol. 11, no. 18, pp. 2801.

"Bottom-UP assembly of nanorobots: extending synthetic biology to complex material design,” Frontiers in Nanoscience and Nanotechnology,vol.5,pp.1-2,2019.

“Re- routing drugs to blood brain barrier: A comprehensive analysis of Machine Learning approaches with fingerprint amalgamation and data balancing,” IEEE Access, vol. 11, pp. 9890-9906, 2023.

“Perspectives on the technological aspects and biomedical applications of Virus-like-particles/ Nanoparticles in reproductive biology: Insights on the medicinal and toxicological outlook,” Advanced NanoBiomed Research, Wiley, 2:2200010, 2022.

"Synergistic and Additive Effects of Menadione in Combination with Antibiotics on Multidrug-Resistant Staphylococcus aureus: Insights from Structure-Function Analysis of Naphthoquinones," ChemMedChem, Chemistry Eurpoe vol. 18, no. 24, 2023.

"Leveraging hallmark Alzheimer’s molecular targets using phytoconstituents: Current perspective and emerging trends," Biomedicine & Pharmacotherapy, Elsevier, vol. 139, no. 111634.

12. Please include the potential limitations of the study.

Reviewer #3: The manuscript titled " Gain efficiency with streamlined and automated data processing: Examples from high-throughput monoclonal antibody production" follows a logical flow of information extending to pipeline design in monoclonal antibody production. The manuscript provides a proof-of concept workflows which is highly interesting. Few minor concerns needs to be addressed.

1. The novelty of the work could be highlighted more clearly. While similar data management systems exist, it's not always obvious what sets this one apart or what new methods it introduces. Comparing its performance and innovations with previous systems would help showcase its distinctive strengths.

2. while the practical challenges of data management in mAB production are provided, the workdesign does not necessarily address these challenges. The workflow design presented is robust but doesn't push the boundaries of what's known in the field of laboratory automation.

3. The manuscript would benefit from referencing more recent advancements in automation, particularly in data management. For example, tools leveraging artificial intelligence and machine learning to optimize workflows, predict outcomes, or detect anomalies in real-time are becoming increasingly relevant in biological data processing.

4. There is a slight confusion when process automation is also highlighted in some places like pipetting schemes etc. This makes it confusing when data mangement is the priority or process automation.

**Do you want your identity to be public for this peer review?** For information about this choice, including consent withdrawal, please see our Privacy Policy

Reviewer #1: **Yes: ** Dr. Priyanka

Reviewer #2: No

Reviewer #3: No

---

## [Author Response · Author response to Decision Letter 1]

3 Feb 2025

In the following, we list all comments and detail how we addressed them.

Journal requirements

Response:

We have adapted the manuscript to meet PLOS ONE’s style requirements.

Response:

All processed data and code are available in the following repositories:

DOI: 10.5281/zenodo.8229164 and DOI: 10.5281/zenodo.10106688

Links to specifically mentioned code are indicated throughout the main text. All three reviewers were satisfied with data availability.

3. Please update your submission to use the PLOS LaTeX template. The template and more information on our requirements for LaTeX submissions can be found at

http://journals.plos.org/plosone/s/latex.

Response:

Thank you for offering to submit with aid of the PLOS LaTeX template. As we understand from PLOS ONE submission guidelines, LaTeX is not mandatory, and manuscripts can also be submitted as .docx files. The latter is preferred by us, indeed for the version with tracked changes.

4. We notice that your supplementary [figures/tables] are included in the manuscript file. Please remove them and upload them with the file type 'Supporting Information'. Please ensure that each Supporting Information file has a legend listed in the manuscript after the references list.

Response:

The supplementary information has been removed from the main text and included as separate files. Each supporting information file has been linked to its legend, as required.

Editor Comments

-The authors should thoroughly address all the major comments from the reviewers and incorporate additional citations if relevant.

-One or more of the reviewers has recommended that you cite specific previously published works. Members of the editorial team have determined that the works referenced are not directly related to the submitted manuscript. As such, please note that it is not necessary or expected to cite the works requested by the reviewer.

Response:

We have assessed all comments from the reviewers and have significantly improved our manuscript. Additional citations relevant to our work have been included, as outlined in the specific responses below.

Reviewer #1:

This article is very good for publication, and the authors have put forth their best effort in writing and presenting the conclusion. However, some changes are required before its publication. I have mentioned all the points below:

I suggest authors to read and incorporate the information from the following articles and cite them:

• mRNA vaccines as an armor to combat the infectious diseases. Travel Medicine and Infectious Disease 52:102550.

• Zoonotic diseases in a changing climate scenario: Revisiting the interplay between environmental variables and infectious disease dynamics, Travel Medicine and Infectious Disease, 58:102694.

• Nanovaccines: A game changing approach in the fight against infectious diseases. Biomedicine & Pharmacotherapy 167(2023):115597

Response: We thank reviewer #1 for the positive evaluation of our manuscript. We have included some of the indicated references that fit into the scope of the manuscript (references below). This adds value to the work and shows the possibility to apply such workflows as ours in different scientific fields.

References:

Priyanka, Abusalah MAH, Chopra H, Sharma A, Mustafa SA, Choudhary OP, et al. Nanovaccines: A game changing approach in the fight against infectious diseases. Biomedicine & Pharmacotherapy. 2023 Nov 1;167:115597. Available from: https://www.sciencedirect.com/science/article/pii/S0753332223013951

Reviewer #2:

The paper has addressed a nice problem; however, I have the below comments:

Response: We thank Reviewer #2 for taking the time to evaluate our manuscript and for providing valuable comments. Many of the challenges highlighted in the review align with those we encounter in our daily work, and we appreciate the opportunity to discuss them further. However, we would like to respectfully elaborate that the primary focus of our manuscript is on data and metadata handling for repetitively executed, complex biological workflows, such as high-throughput monoclonal antibody cloning and in vitro production.

1. Accurately isolating individual B-cells from heterogeneous populations can be technically demanding. Techniques like flow cytometry or microfluidics need to be highly precise to ensure that only the target B-cell is selected.

Response: The biological protocol we are using is based on flow cytometry as described previously in the works cited by Tiller T. et al., 2008 and Kreye J. et al., 2016. With regard to our approach to data and metadata tracking, as described in our manuscript, it is important to track key metadata such as FACS machine type and the key settings used, along with the utilized cell markers etc.

2. Maintaining the viability of single B-cells during isolation and subsequent culture is crucial. Many methods can stress or damage cells, reducing their ability to produce antibodies. Please cite the below papers while discussing this:

"Integrative Toxicogenomics: Advancing Precision Medicine and Toxicology through Artificial Intelligence and OMICs Technology," Biomedicine & Pharmacotherapy, Elsevier, vol. 163, 114784, 2023.

“Advances in smoking related in-vitro inhalation toxicology: a perspective case of challenges and opportunities from progresses in lung-on-chip technologies,” ACS Chemical Research in Toxicology, vol. 34, pp. 1984-222, 2021.

“Emerging technologies for in vitro inhalation toxicology,” Advanced Healthcare Materials, Wiley, vol. 10, no. 18, pp. e2100633 2021.

Response: We have included the below-mentioned references to further explain the application of our workflow and have expanded our discussion on the role of AI and ML in design principles, data processing and analyses.

References:

Singh AV, Chandrasekar V, Paudel N, Laux P, Luch A, Gemmati D, et al. Integrative toxicogenomics: Advancing precision medicine and toxicology through artificial intelligence and OMICs technology. Biomed Pharmacother [Internet]. 2023;163:114784. Available from: http://dx.doi.org/10.1016/j.biopha.2023.114784

Singh AV, Romeo A, Scott K, Wagener S, Leibrock L, Laux P, et al. Emerging technologies for in vitro inhalation toxicology. Adv Healthc Mater [Internet]. 2021;10(18). Available from: http://dx.doi.org/10.1002/adhm.202100633

Concerning the maintenance of viability during B-cell isolation and culture, we do agree that this is a stressful process for the cells and requires careful handling and protocol optimization. However, for the purpose of monoclonal antibody generation, we do not culture the B-cells after FACS sorting, rather directly lyse them and isolate RNA that is in turn translated to cDNA.

3. Once isolated, the B-cells must be effectively expanded to produce sufficient quantities of antibodies. Achieving robust growth while preserving their functionality can be difficult.

Response: For protocols utilizing B-cell culture to produce antibodies, achieving B-cell expansion while preserving functionality is a considerable challenge. While our manuscript primarily focuses on data handling, the biological workflow underlying the data pipeline uses a different strategy for antibody production. Instead of culturing B-cells, we directly isolate RNA to clone the antibodies (see also above).

4. Not all B-cells will produce high-affinity or functional antibodies. Identifying and selecting the right clones that produce the desired antibodies requires additional screening and validation steps.

Response: Absolutely! While our focus is on antibody production, B-cell characterization presents another challenge that can be addressed through various techniques, each with its own advantages and limitations. See for example: Kreye J, Reincke SM, Kornau H, Sánchez-Sendin E, Corman VM, Liu H, et al. A Therapeutic Non-self-reactive SARS-CoV-2 Antibody Protects from Lung Pathology in a COVID-19 Hamster Model. Cell. 2020 Nov;183(4):1058-1069.e19.

5. The genetic material of isolated B-cells can be unstable, leading to mutations or loss of antibody specificity during culture. Ensuring genetic integrity is crucial for consistent antibody production.

Response: The risk of genetic instability increases with extended culturing time, as observed, for example, in hybridoma technology. To mitigate this risk, we avoid B cell culturing. Instead, we directly clone antibodies from individual B-cells, adhering strictly to best practices in RNA and DNA handling.

6. Traditional methods of creating monoclonal antibodies often involve hybridoma technology, which adds complexity and can be less efficient compared to newer single-cell approaches.

Response: We completely agree. Hybridoma technology has been the gold standard for decades, offering advantages such as longevity and the ability to conduct extended phenotypic screening. However, modern methods, such as the one we employ, are faster, offer higher specificity, and ensure genetic accuracy.

7. Scaling up production from a single B-cell to large-scale antibody production can present logistical and technical challenges, including maintaining consistency and quality across batches.

Response: The most efficient and controllable approach for large-scale antibody production is in vitro production. Our focus is on the generation of plasmids designed for cell transfection to facilitate in vitro antibody production. While our plasmids enable the production of large quantities of a single monoclonal antibody, we specialize in cloning hundreds to thousands of distinct monoclonal antibodies.

8. Automating the entire process—from isolation to production—while maintaining high throughput and reproducibility can be complex and resource-intensive.

Response: We automated the entire process, from cDNA generation to antibody production. Establishing this workflow is indeed resource-intensive, with significant organizational complexity. The key to reducing complexity and improving efficiency during routing operation lies in automated data handling, which has been a central focus of our efforts. Our motivation to share these learnings is reflected in the presented manuscript, and we further emphasize this point in the revised version.

9. The process of characterizing the antibodies produced (e.g., affinity, specificity) can be time-consuming and requires sophisticated assays, adding to the overall complexity.

Response: The first step to further characterize monoclonal antibodies is their production. Antibody cloning from individual B-cells for in vitro production represents the most effective approach to generate enough antibody for all required assays. We demonstrate that this production process is feasible in a highly automated manner, enabling the generation of over 1,000 different antibodies per year.

10. Finally, the paper should be thoroughly proofread and written like a scientific manuscript.

Response: The manuscript has been proofread and adapted to fit the scientific manuscript requirements of PLOS ONE.

11. Please discuss the role of permeability because it plays a crucial role in the complex production of monoclonal antibodies from single B-cells: High permeability of the cell membrane is essential for the efficient exchange of nutrients and oxygen. This ensures that the isolated B-cells receive the necessary metabolic support for growth and antibody production.

Please cite the below papers:

“Investigating the Use of Machine Learning Models to Understand the Drugs Permeability Across Placenta,” IEEE Access, vol. 11, pp. 52726-52739, 2023. “Micropatterned neurovascular interface to mimic the blood-brain barrier neurophysiology and micromechanical function,” Cells, MDPI, vol. 11, no. 18, pp. 2801. "Bottom-UP assembly of nanorobots: extending synthetic biology to complex material design,” Frontiers in Nanoscience and Nanotechnology, vol.5,pp.1-2,2019. “Re- routing drugs to blood brain barrier: A comprehensive analysis of Machine Learning approaches with fingerprint amalgamation and data balancing,” IEEE Access, vol. 11, pp. 9890-9906, 2023. “Perspectives on the technological aspects and biomedical applications of Virus-like-particles/ Nanoparticles in reproductive biology: Insights on the medicinal and toxicological outlook,” Advanced NanoBiomed Research, Wiley, 2:2200010, 2022. "Synergistic and Additive Effects of Menadione in Combination with Antibiotics on Multidrug-Resistant Staphylococcus aureus: Insights from Structure-Function Analysis of Naphthoquinones," ChemMedChem, Chemistry Eurpoe vol. 18, no. 24, 2023. "Leveraging hallmark Alzheimer’s molecular targets using phytoconstituents: Current perspective and emerging trends," Biomedicine & Pharmacotherapy, Elsevier,vol. 139, no. 111634.

Response: Permeability and the regulation of permeability across barriers are essential for the compartmentalized functions of cells and tissues. This is true not only for cultured B-cells but also HEK cells, which we use for overexpressing monoclonal antibodies. In the context of our manuscript, permeability is particularly significant in automated cell culture to produce brain microvascular endothelial cells for our blood-brain barrier model. We have included below mentioned references and further strengthened our discussion regarding this point.

References added:

Ansari MY, Chandrasekar V, Singh AV, Dakua SP. Re-routing drugs to blood brain barrier: A comprehensive analysis of machine learning approaches with fingerprint amalgamation and data balancing. IEEE Access [Internet]. 2023;11:9890–906. Available from: http://dx.doi.org/10.1109/access.2022.3233110

Dhage PA, Sharbidre AA, Dakua SP, Balakrishnan S. Leveraging hallmark Alzheimer’s molecular targets using phytoconstituents: Current perspective and emerging trends. Biomed Pharmacother [Internet]. 2021;139:111634. Available from: http://dx.doi.org/10.1016/j.biopha.2021.11163

12. Please include the potential limitations of the study.

Response: As any other study dealing with data processing, this too has limitations that need to be considered. To highlight these, we have added a separate “Limitations” paragraph in the main text and discussed several points, including how to potentially improve and/or solve them for future workflows. We hope that this will be helpful for future studies and laboratories dealing with automated data processing and database generation.

Reviewer #3:

The manuscript titled " Gain efficiency with streamlined and automated data processing: Examples from high-throughput monoclonal antibody production" follows a logical flow of information extending to pipeline design in monoclonal antibody production. The manuscript provides a proof-of concept workflows which is highly interesting. Few minor concerns needs to be addressed.

1. The novelty of the work could be highlighted more clearly. While similar data management systems exist, it's not always obvious what sets this one apart or what new methods it introduces. Comparing its per

---

## [Decision Letter · Decision Letter 1]

18 Mar 2025

Dear Dr. Stappert,

**Authors are advised to revise the manuscript according to the comments provided by Reviewer #4.**

We look forward to receiving your revised manuscript.

Kind regards,

Bhanwar Lal Puniya, Ph.D.

Academic Editor

PLOS ONE

Reviewers' comments:

Reviewer's Responses to Questions

**Comments to the Author**

Reviewer #3: All comments have been addressed

Reviewer #4: (No Response)

2. Is the manuscript technically sound, and do the data support the conclusions?

Reviewer #3: Yes

Reviewer #4: (No Response)

3. Has the statistical analysis been performed appropriately and rigorously?

Reviewer #3: Yes

Reviewer #4: (No Response)

4. Have the authors made all data underlying the findings in their manuscript fully available?

Reviewer #3: Yes

Reviewer #4: (No Response)

5. Is the manuscript presented in an intelligible fashion and written in standard English?

Reviewer #3: Yes

Reviewer #4: (No Response)

**Reviewer #3:**  (No Response)

**Reviewer #4: ** I would like to thank the authors for their efforts in preparing this manuscript. It presents an important contribution, please addressed below comments :

• Some sections feel a bit dense and could be made clearer by breaking up longer sentences and simplifying complex descriptions. This would help improve readability and accessibility, especially for readers who may not be familiar with all the technical details.

• The transitions between sections, particularly from general workflow principles to specific implementation details, could be smoother. A clearer connection between these parts would help guide the reader more effectively.

• Figure 2 seems to be an important part of explaining the workflow, but it’s not included for review. Make sure it’s well-labeled and clearly conveys the intended message.

• It would also help to refer to workflow diagrams more explicitly in the text so readers can follow along more easily.

• The manuscript briefly mentions why Python and Knime were chosen, but it would be useful to provide a stronger rationale. A short comparison of the benefits and trade-offs of these tools versus other available options would add clarity.

• Similarly, when discussing GUI-based vs. scripting-based approaches, consider touching on usability, performance, and reproducibility. This would give readers a better understanding of the reason behind these choices.

• The paper mentions input and output file formats but doesn’t fully address the importance of standardization for interoperability. A brief mention of commonly used formats (e.g., CSV, JSON, XML) and their advantages would help strengthen this section.

• It would also be helpful to explain how intermediate files are validated to ensure data integrity as information moves between pipeline modules.

• The discussion on computational resources (e.g., NAS, cloud services, institutional servers) could benefit from a short mention of data security and access control measures. This would be especially relevant for handling sensitive or large-scale datasets.

• Since working with large datasets can present challenges, consider addressing potential issues like memory constraints or options for parallel computing.

**Do you want your identity to be public for this peer review?** For information about this choice, including consent withdrawal, please see our Privacy Policy

Reviewer #3: No

Reviewer #4: No

---

## [Author Response · Author response to Decision Letter 2]

8 May 2025

Authors are advised to revise the manuscript according to the comments provided by Reviewer #4.

We have addressed all comments and suggestions provided by Reviewer #4. We are thankful for the critical review process and hope that the improved manuscript now fully meets the publication criteria of PLOS ONE.

Reviewer #4:

I would like to thank the authors for their efforts in preparing this manuscript. It presents an important contribution, please addressed below comments:

We thank Reviewer #4 for the critical evaluation of our manuscript and for recognizing its importance. We have addressed all comments in the revised version and expanded on those in our rebuttal letter.

1. Some sections feel a bit dense and could be made clearer by breaking up longer sentences and simplifying complex descriptions. This would help improve readability and accessibility, especially for readers who may not be familiar with all the technical details. (answered together with comment no. 2, below comment no. 2)

2. The transitions between sections, particularly from general workflow principles to specific implementation details, could be smoother. A clearer connection between these parts would help guide the reader more effectively.

We appreciate these valuable suggestions. In response, we have revised the main text by breaking up longer sentences and making technical details easier to follow and read. Complex descriptions have been simplified where appropriate, without compromising the essence and structure of the manuscript. We have also tried to establish clear links between general and technical details, especially in our ‘Examples from the workflow’ sections. This also helps to follow the logical structure of the manuscript better. Our Glossary (Box 1) further aids readers at the start by defining data processing concepts often used throughout the text (refer to page 5) and was also refined to convey clearer messages in shorter sentences.

We think that the manuscript is now much more coherent and accessible.

3. Figure 2 seems to be an important part of explaining the workflow, but it’s not included for review. Make sure it’s well-labeled and clearly conveys the intended message.

Figure 2 is indeed an essential part of our manuscript, illustrating the entire mAB production pipeline and the many different steps involved in a logical order. Figure 2 has been recalled multiple times throughout the text, pointing out examples of how our workflow is designed. We have made sure that the figure is explained in the manuscript and that it is properly referred to in each section where appropriate. We have also labelled and annotated it. We will of course include it with our revision file (see also below).

Fig 2. Workflow for the production of mABs from patient-derived individual B-cells. Wet lab work steps (ellipse-icons with headers) and data processing steps (yellow icons). Sample flow is depicted by blue arrows and information flow is depicted by yellow arrows. The experimental procedure starts with FACS sorting of individual B-cells. cDNA is generated from individual B-cells, and PCR1 and PCR2 serve to amplify antibody chains. Successfully amplified chains, evaluated by capillary electrophoresis (cELE1), are sequenced (SEQ1). Upon positive sequence evaluation with the aBASE software [23], specific primers are used for the final amplification of the specificity-determining chain parts (PCR3). Amplification is quality-controlled by electrophoresis (cELE2). The specificity-determining regions are cloned by Gibson Assembly (GiAs) into plasmids encoding the constant parts of the chains. Plasmids are amplified by transformed (TRFO) bacteria after plating (PLA) and picking (PICK) individual bacterial clones. Prior to quality control by sequencing (SEQ2), amplified plasmids are isolated (MINI), and plasmid concentration is measured and adjusted (ConcAdj). Based on sequencing results, functional plasmids (PLA) are sorted. Matching plasmid pairs (mAB Pla) are prepared for transfection (Traf) into HEK cells. mABs produced and secreted by HEK cells are harvested (mABs), and mAB concentration is quantified (quant). Glycerol stocks (Glyc) of transformed bacteria and plasmid aliquots (Pla safe) serve as safe stocks of plasmids. The colors of the wet lab work step captions indicate different workflow sections: blue – molecular biology; red – microbiology; green – cell biology. The colors of the data processing circle contours and font indicate the utilized technology: red – Python; yellow – Knime; blue – third-party software.

4. It would also help to refer to workflow diagrams more explicitly in the text so readers can follow along more easily.

We agree with this comment and have made sure that each workflow diagram is explicitly referred to at relevant points in-text, helping guide readers through the process more clearly.

5. The manuscript briefly mentions why Python and Knime were chosen, but it would be useful to provide a stronger rationale. A short comparison of the benefits and trade-offs of these tools versus other available options would add clarity.

Thank you for the suggestion. We have expanded upon the explanation of our choice of Python and KNIME, emphasizing their complementary advantages (please see also our answer to comment no. 6 below)

6. Similarly, when discussing GUI-based vs. scripting-based approaches, consider touching on usability, performance, and reproducibility. This would give readers a better understanding of the reason behind these choices.

We appreciate the suggestion to elaborate further on our selection of tools, GUI-based and script-based approaches.

We have added a brief description of why Knime and Python were chosen, as well as some suggestions for alternative technologies (refer to “Examples from the workflow” on page 11). We have also expanded on the usage of GUI-based workflows (page 10).

Knime was chosen as a technology mainly due to its wide application and open-source system. Knime has an intuitive drag-and-drop graphical interface and can easily be used by team members without coding experience. Furthermore, workflows can be shared, ensuring reproducibility and flexibility. Knime can also embed Python scripts.

Python is very widespread in the field of data processing and bioinformatics. It is suitable for many different projects and computations. It can run on various operating systems and supports notebook-style reporting, further enhancing usability and reproducibility of workflows. The syntax used by Python is generally clean and readable, which is helpful when working with interdisciplinary teams.

The use of GUI tools in our workflow is essential to allow seamless and faster integration of researchers with limited coding knowledge on an existing pipeline. The combined use of tools allows interdisciplinary work and ensures consistency, standardized documentation, and automation of repetitive tasks.

7. The paper mentions input and output file formats but doesn’t fully address the importance of standardization for interoperability. A brief mention of commonly used formats (e.g., CSV, JSON, XML) and their advantages would help strengthen this section.

We agree that the use of standardized files plays a key role in interoperability and reproducibility across platforms. Our pipeline supports CSV files which are commonly used in laboratory settings. CSV files are easy to read, simple and compatible with spreadsheet software and most programming environments. We have clarified the use of CSV files in our manuscript (page 15). Formats such as JSON or XML support much larger nested data and metadata which we do not handle in this pipeline. Our datasets are primarily tabular, as are most datasets generated in labs. Nevertheless, we added a brief section on the context in which JSON and XML data formats could be used.

8. It would also be helpful to explain how intermediate files are validated to ensure data integrity as information moves between pipeline modules.

Thank you for the comment. Indeed, this section was lacking clarification of that point. We have added a description of our built-in validation steps for intermediate files, please refer to the box “Examples from the workflow” in the section “Defining input and output files”.

9. The discussion on computational resources (e.g., NAS, cloud services, institutional servers) could benefit from a short mention of data security and access control measures. This would be especially relevant for handling sensitive or large-scale datasets.

Thank you for the comment. We have expanded the discussion on computational resources to emphasize that, when using storage drives and servers such as NAS, cloud services, or institutional infrastructure, appropriate data security and access control measures should be in place—particularly when managing sensitive or large-scale datasets. This includes implementing practices such as user authentication, encryption, and compliance with institutional or legal data protection guidelines.

10. Since working with large datasets can present challenges, consider addressing potential issues like memory constraints or options for parallel computing.

Thank you for the suggestion, we do agree it is important to address these, especially considering ever-increasing volume of (biological) data. We have added a sentence acknowledging the challenges of working with large datasets and briefly discussed strategies such as memory management and parallel processing. Please refer to the section” Allocating separate computational space to modules” on page 14 for revision of these changes.

We look forward to your response.

Best Regards,

Dominik Stappert

---

## [Decision Letter · Decision Letter 2]

30 May 2025

Gain efficiency with streamlined and automated data processing: Examples from high-throughput monoclonal antibody production

PONE-D-24-34015R2

Dear Dr. Stappert,

We’re pleased to inform you that your manuscript has been judged scientifically suitable for publication and will be formally accepted for publication once it meets all outstanding technical requirements.

Kind regards,

Bhanwar Lal Puniya, Ph.D.

Academic Editor

PLOS ONE

Additional Editor Comments (optional):

Reviewers' comments:

Reviewer's Responses to Questions

**Comments to the Author**

Reviewer #4: (No Response)

2. Is the manuscript technically sound, and do the data support the conclusions?

Reviewer #4: (No Response)

3. Has the statistical analysis been performed appropriately and rigorously?

Reviewer #4: (No Response)

4. Have the authors made all data underlying the findings in their manuscript fully available?

Reviewer #4: (No Response)

5. Is the manuscript presented in an intelligible fashion and written in standard English?

Reviewer #4: (No Response)

Reviewer #4: (No Response)

**Do you want your identity to be public for this peer review?** For information about this choice, including consent withdrawal, please see our Privacy Policy

Reviewer #4: No

---

## [Editor Report · Acceptance letter]

PONE-D-24-34015R2

PLOS ONE

Dear Dr. Stappert,

I'm pleased to inform you that your manuscript has been deemed suitable for publication in PLOS ONE. Congratulations! Your manuscript is now being handed over to our production team.

Kind regards,

on behalf of

Dr. Bhanwar Lal Puniya

Academic Editor

PLOS ONE